# Manifold-tiling Localized Receptive Fields are Optimal in Similarity-preserving Neural Networks

**Anirvan M. Sengupta**[†‡]   **Mariano Tepper**[‡*]   **Cengiz Pehlevan**[‡*]
**Alexander Genkin**[§]   **Dmitri B. Chklovskii**[‡§]

[†]Rutgers University   [‡]Flatiron Institute   [§]NYU Langone Medical Center
anirvans@physics.rutgers.edu, alexander.genkin@gmail.com
{mtepper,cpehlevan,dchklovskii}@flatironinstitute.org

## Abstract

Many neurons in the brain, such as place cells in the rodent hippocampus, have localized receptive fields, i.e., they respond to a small neighborhood of stimulus space. What is the functional significance of such representations and how can they arise? Here, we propose that localized receptive fields emerge in similarity-preserving networks of rectifying neurons that learn low-dimensional manifolds populated by sensory inputs. Numerical simulations of such networks on standard datasets yield manifold-tiling localized receptive fields. More generally, we show analytically that, for data lying on symmetric manifolds, optimal solutions of objectives, from which similarity-preserving networks are derived, have localized receptive fields. Therefore, nonnegative similarity-preserving mapping (NSM) implemented by neural networks can model representations of continuous manifolds in the brain.

## 1 Introduction

A salient and unexplained feature of many neurons is that their receptive fields are localized in the parameter space they represent. For example, a hippocampus place cell is active in a particular spatial location [1], the response of a V1 neuron is localized in visual space and orientation [2], and the response of an auditory neuron is localized in the sound frequency space [3]. In all these examples, receptive fields of neurons from the same brain area tile (with overlap) low-dimensional manifolds.

Localized receptive fields are shaped by neural activity as evidenced by experimental manipulations in developing and adult animals [4, 5, 6, 7]. Activity influences receptive fields via modification, or learning, of synaptic weights which gate the activity of upstream neurons channeling sensory inputs. To be biologically plausible, synaptic learning rules must be physically local, i.e., the weight of a synapse depends on the activity of only the two neurons it connects, pre- and post-synaptic.

In this paper, we demonstrate that biologically plausible neural networks can learn manifold-tiling localized receptive fields from the upstream activity in an unsupervised fashion. Because analyzing the outcome of learning in arbitrary neural networks is often difficult, we take a normative approach, Fig. 1. First, we formulate an optimization problem by postulating an objective function and constraints, Fig. 1. Second, for inputs lying on a manifold, we derive an optimal offline solution and demonstrate analytically and numerically that the receptive fields are localized and tile the manifold, Fig. 1. Third, from the same objective, we derive an online optimization algorithm which can be implemented by a biologically plausible neural network, Fig. 1. We expect this network to learn localized receptive fields, the conjecture we confirm by simulating the network numerically, Fig. 1.

Optimization functions considered here belong to the family of similarity-preserving objectives which dictate that similar inputs to the network elicit similar outputs [8, 9, 10, 11, 12]. In the absence of sign

---

[*]M. Tepper and C. Pehlevan contributed equally to this work.

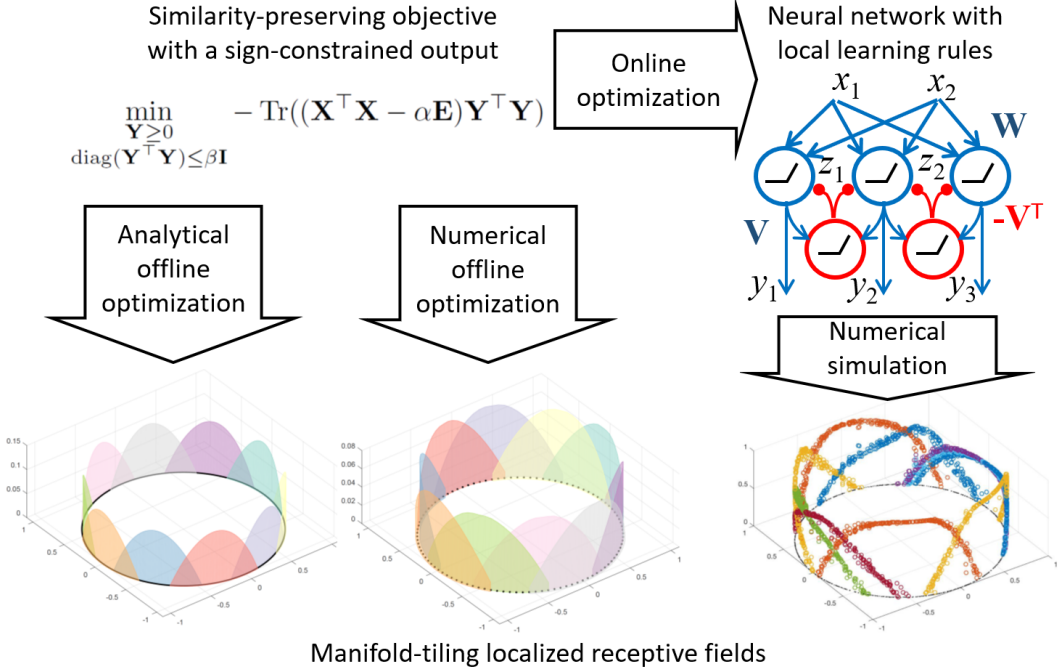

$$\min_{\substack{\mathbf{Y} \geq 0 \\ \mathrm{diag}(\mathbf{Y}^\top \mathbf{Y}) \leq \beta \mathbf{I}}} - \mathrm{Tr}((\mathbf{X}^\top \mathbf{X} - \alpha \mathbf{E})\mathbf{Y}^\top \mathbf{Y})$$

Similarity-preserving objective with a sign-constrained output

Online optimization

Neural network with local learning rules

Analytical offline optimization

Numerical offline optimization

Numerical simulation

Manifold-tiling localized receptive fields

Figure 1: A schematic illustration of our normative approach.

constraints, such objectives are provably optimized by projecting inputs onto the principal subspace [13, 14, 15], which can be done online by networks of linear neurons [8, 9, 10]. Constraining the sign of the output leads to networks of rectifying neurons [11] which have been simulated numerically in the context of clustering and feature learning [11, 12, 16, 17], and analyzed in the context of blind source extraction [18]. In the context of manifold learning, optimal solutions of **N**onnegative **S**imilarity-preserving **M**apping objectives have been missing because optimization of existing NSM objectives is challenging. Our main contributions are:

- Analytical optimization of NSM objectives for input originating from symmetric manifolds.
- Derivation of biologically plausible NSM neural networks.
- Offline and online algorithms for manifold learning of arbitrary manifolds.

The paper is organized as follows. In Sec. 2, we derive a simplified version of an NSM objective. Much of our following analysis can be carried over to other NSM objectives but with additional technical considerations. In Sec. 3, we derive a necessary condition for the optimal solution. In Sec. 4, we consider solutions for the case of symmetric manifolds. In Sec. 5, we derive online optimization algorithm and an NSM neural network. In Sec. 6, we present the results of numerical experiments, which can be reproduced with the code at `https://github.com/flatironinstitute/mantis`.

## 2   A Simplified Similarity-preserving Objective Function

To introduce similarity-preserving objectives, let us define our notation. The input to the network is a set of vectors, $\mathbf{x}_t \in \mathbb{R}^n, t = 1, \ldots, T$, with components represented by the activity of $n$ upstream neurons at time, $t$. In response, the network outputs an activity vector, $\mathbf{y}_t \in \mathbb{R}^m, t = 1, \ldots, T$, $m$ being the number of output neurons.

Similarity preservation postulates that similar input pairs, $\mathbf{x}_t$ and $\mathbf{x}_{t'}$, evoke similar output pairs, $\mathbf{y}_t$ and $\mathbf{y}_{t'}$. If similarity of a pair of vectors is quantified by their scalar product and the distance metric of similarity is Euclidean, we have

$$\min_{\forall t \in \{1,\ldots,T\}: \mathbf{y}_t \in \mathbb{R}^m} \frac{1}{2} \sum_{t,t'=1}^{T} (\mathbf{x}_t \cdot \mathbf{x}_{t'} - \mathbf{y}_t \cdot \mathbf{y}_{t'})^2 = \min_{\mathbf{Y} \in \mathbb{R}^{m \times T}} \frac{1}{2} \|\mathbf{X}^\top \mathbf{X} - \mathbf{Y}^\top \mathbf{Y}\|_F^2, \qquad (1)$$

where we introduced a matrix notation $\mathbf{X} \equiv [\mathbf{x}_1, \ldots, \mathbf{x}_T] \in \mathbb{R}^{n \times T}$ and $\mathbf{Y} \equiv [\mathbf{y}_1, \ldots, \mathbf{y}_T] \in \mathbb{R}^{m \times T}$ and $m < n$. Such optimization problem is solved offline by projecting the input data to the principal subspace [13, 14, 15]. The same problem can be solved online by a biologically plausible neural network performing global linear dimensionality reduction [8, 10].

We will see below that nonlinear manifolds can be learned by constraining the sign of the output and introducing a similarity threshold, $\alpha$ (here $\mathbf{E}$ is a matrix of all ones):

$$\min_{\mathbf{Y} \geq 0} \tfrac{1}{2} \|\mathbf{X}^\top \mathbf{X} - \alpha \mathbf{E} - \mathbf{Y}^\top \mathbf{Y}\|_F^2 = \min_{\forall t:\, \mathbf{y}_t \geq 0} \tfrac{1}{2} \sum_{t,t'} (\mathbf{x}_t \cdot \mathbf{x}_{t'} - \alpha - \mathbf{y}_t \cdot \mathbf{y}_{t'})^2, \tag{2}$$

In the special case, $\alpha = 0$, Eq. (2) reduces to the objective in [11, 19, 18].

Intuitively, Eq. (2) attempts to preserve similarity for similar pairs of input samples but orthogonalizes the outputs corresponding to dissimilar input pairs. Indeed, if the input similarity of a pair of samples $t, t'$ is above a specified threshold, $\mathbf{x}_t \cdot \mathbf{x}_{t'} > \alpha$, output vectors $\mathbf{y}_t$ and $\mathbf{y}_{t'}$ would prefer to have $\mathbf{y}_t \cdot \mathbf{y}_{t'} \approx \mathbf{x}_t \cdot \mathbf{x}_{t'} - \alpha$, i.e., it would be similar. If, however, $\mathbf{x}_t \cdot \mathbf{x}_{t'} < \alpha$, the lowest value $\mathbf{y}_t \cdot \mathbf{y}_{t'}$ for $\mathbf{y}_t, \mathbf{y}_{t'} \geq 0$ is zero meaning that that they would tend to be orthogonal, $\mathbf{y}_t \cdot \mathbf{y}_{t'} = 0$. As $\mathbf{y}_t$ and $\mathbf{y}_{t'}$ are nonnegative, to achieve orthogonality, the output activity patterns for dissimilar patterns would have non-overlapping sets of active neurons. In the context of manifold representation, Eq. (2) strives to preserve in the $\mathbf{y}$-representation local geometry of the input data cloud in $\mathbf{x}$-space and let the global geometry emerge out of the nonlinear optimization process.

As the difficulty in analyzing Eq. (2) is due to the quartic in $\mathbf{Y}$ term, we go on to derive a simpler quadratic in $\mathbf{Y}$ objective function that produces very similar outcomes. To this end, we, first, introduce an additional power constraint: $\operatorname{Tr} \mathbf{Y}^\top \mathbf{Y} \leq k$ as in [9, 11]. We will call the input-output mapping obtained by this procedure NSM-0:

$$\operatorname*{argmin}_{\substack{\mathbf{Y} \geq 0 \\ \operatorname{Tr} \mathbf{Y}^\top \mathbf{Y} \leq k}} \frac{1}{2} \|\mathbf{X}^\top \mathbf{X} - \alpha \mathbf{E} - \mathbf{Y}^\top \mathbf{Y}\|_F^2 = \operatorname*{argmin}_{\substack{\mathbf{Y} \geq 0 \\ \operatorname{Tr} \mathbf{Y}^\top \mathbf{Y} \leq k}} - \operatorname{Tr}((\mathbf{X}^\top \mathbf{X} - \alpha \mathbf{E}) \mathbf{Y}^\top \mathbf{Y}) + \frac{1}{2} \|\mathbf{Y}^\top \mathbf{Y}\|_F^2,$$
$$\text{(NSM-0)}$$

where we expanded the square and kept only the $\mathbf{Y}$-dependent terms.

We can redefine the variables and drop the last term in a certain limit (see the Supplementary Material, Sec. A.1, for details) leading to the optimization problem we call NSM-1:

$$\min_{\substack{\mathbf{Y} \geq 0 \\ \operatorname{diag}(\mathbf{Y}^\top \mathbf{Y}) \leq \beta \mathbf{I}}} - \operatorname{Tr}((\mathbf{X}^\top \mathbf{X} - \alpha \mathbf{E}) \mathbf{Y}^\top \mathbf{Y}) = \min_{\substack{\forall t \in \{1, \ldots, T\}: \\ \mathbf{y}_t \geq 0,\, \|\mathbf{y}_t\|_2^2 \leq \beta}} - \sum_{t,t'} (\mathbf{x}_t \cdot \mathbf{x}_{t'} - \alpha) \mathbf{y}_t \cdot \mathbf{y}_{t'}. \quad \text{(NSM-1)}$$

Conceptually, this type of objective has proven successful for manifold learning [20]. Intuitively, just like Eq. (2), NSM-1 preserves similarity of nearby input data samples while orthogonalizing output vectors of dissimilar input pairs. Indeed, a pair of samples $t, t'$ with $\mathbf{x}_t \cdot \mathbf{x}_{t'} > \alpha$, would tend to have $\mathbf{y}_t \cdot \mathbf{y}_{t'}$ as large as possible, albeit with the norm of the vectors controlled by the constraint $\|\mathbf{y}_t\|^2 \leq \beta$. Therefore, when the input similarity for the pair is above a specified threshold, the vectors $\mathbf{y}_t$ and $\mathbf{y}_{t'}$ would prefer to be aligned in the same direction. For dissimilar inputs with $\mathbf{x}_t \cdot \mathbf{x}_{t'} < \alpha$, the corresponding output vectors $\mathbf{y}_t$ and $\mathbf{y}_{t'}$ would tend to be orthogonal, meaning that responses to these dissimilar inputs would activate mostly nonoverlapping sets of neurons.

## 3 A Necessary Optimality Condition for NSM-1

In this section, we derive the necessary optimality condition for Problem (NSM-1). For notational convenience, we introduce the Gramian $\mathbf{D} \equiv \mathbf{X}^\top \mathbf{X}$ and use $[\mathbf{z}]_+$, where $\mathbf{z} \in \mathbb{R}^T$, for the component-wise ReLU function, $([\mathbf{z}]_+)_t \equiv \max(z_t, 0)$.

**Proposition 1.** *The optimal solution of Problem (NSM-1) satisfies*

$$[(\mathbf{D} - \alpha \mathbf{E}) \mathbf{y}^{(a)}]_+ = \mathbf{\Lambda} \mathbf{y}^{(a)}, \tag{3}$$

*where $\mathbf{y}^{(a)}$ designates a column vector which is the transpose of the $a$-th row of $\mathbf{Y}$ and $\mathbf{\Lambda} = \operatorname{diag}(\lambda_1, \ldots, \lambda_T)$ is a nonnegative diagonal matrix.*

The proof of Proposition 1 (Supplementary Material, Sec. A.2) proceeds by introducing Lagrange multipliers $\mathbf{\Lambda} = \mathrm{diag}(\lambda_1, \dots, \lambda_T) \geq 0$ for the constraint $\mathrm{diag}(\mathbf{Y}^\top \mathbf{Y}) \leq \beta \mathbf{I}$, and writing down the KKT conditions. Then, by separately considering the cases $\lambda_t y_{at} = 0$ and $\lambda_t y_{at} > 0$ we get Eq. (3).

To gain insight into the nature of the solutions of (3), let us assume $\lambda_t > 0$ for all $t$ and rewrite it as

$$y_{at} = \left[ \frac{1}{\lambda_t} \sum_{t'} (D_{tt'} - \alpha) y_{at'} \right]_+. \tag{4}$$

Eq. (4) suggests that the sign of the interaction within each pair of $\mathbf{y}_t$ and $\mathbf{y}_{t'}$ depends on the similarity of the corresponding inputs. If $\mathbf{x}_t$ and $\mathbf{x}_{t'}$ are similar, $D_{tt'} > \alpha$, then $y_{at'}$ has excitatory influence on $y_{at}$. Otherwise, if $\mathbf{x}_t$ and $\mathbf{x}_{t'}$ are farther apart, the influence is inhibitory. Such models often give rise to localized solutions [21]. Since, in our case, the variable $y_{at}$ gives the activity of the $a$-th neuron as the $t$-th input vector is presented to the network, such a solution would define a receptive field of neuron, $a$, localized in the space of inputs. Below, we will derive such localized-receptive field solutions for inputs originating from symmetric manifolds.

## 4 Solution for Symmetric Manifolds via a Convex Formulation

So far, we set the dimensionality of $\mathbf{y}$, i.e., the number of output neurons, $m$, a priori. However, as this number depends on the dataset, we would like to allow for flexibility of choosing the output dimensionality adaptively. To this end, we introduce the Gramian, $\mathbf{Q} \equiv \mathbf{Y}^\top \mathbf{Y}$, and do not constrain its rank. Minimization of our objective functions requires that the output similarity expressed by Gramian, $\mathbf{Q}$, captures some of the input similarity structure encoded in the input Gramian, $\mathbf{D}$.

Redefining the variables makes the domain of the optimization problem convex. Matrices like $\mathbf{D}$ and $\mathbf{Q}$ which could be expressed as Gramians are symmetric and positive semidefinite. In addition, any matrix, $\mathbf{Q}$, such that $\mathbf{Q} \equiv \mathbf{Y}^\top \mathbf{Y}$ with $\mathbf{Y} \geq 0$ is called *completely positive*. The set of completely positive $T \times T$ matrices is denoted by $\mathcal{CP}^T$ and forms a closed convex cone [22].

Then, NSM-1, without the rank constraint, can be restated as a convex optimization problem with respect to $\mathbf{Q}$ belonging to the convex cone $\mathcal{CP}^T$:

$$\min_{\substack{\mathbf{Q} \in \mathcal{CP}^T \\ \mathrm{diag}(\mathbf{Q}) \leq \beta \mathbf{I}}} - \mathrm{Tr}((\mathbf{D} - \alpha \mathbf{E})\mathbf{Q}). \tag{NSM-1a}$$

Despite the convexity, for arbitrary datasets, optimization problems in $\mathcal{CP}^T$ are often intractable for large $T$ [22]. Yet, for $\mathbf{D}$ with a high degree of symmetry, below, we will find the optimal $\mathbf{Q}$.

Imagine now that there is a group $G \subseteq S_T$, $S_T$ being the permutation group of the set $\{1, 2, \dots, T\}$, so that $D_{g(t)g(t')} = D_{tt'}$ for all $g \in G$. The matrix with elements $M_{g(t)g(t')}$ is denoted as $g\mathbf{M}$, representing group action on $\mathbf{M}$. We will represent action of $g$ on a vector $\mathbf{w} \in R^T$ as $g\mathbf{w}$, with $(g\mathbf{w})_t = w_{g(t)}$.

**Theorem 1.** *If the action of the group $G$ is transitive, that is, for any pair $t, t' \in \{1, 2, \dots, T\}$ there is a $g \in G$ so that $t' = g(t)$, then there is at least one optimal solution of Problem (NSM-1a) with $\mathbf{Q} = \mathbf{Y}^\top \mathbf{Y}, \mathbf{Y} \in \mathbb{R}^{m \times T}$ and $\mathbf{Y} \geq 0$, such that*

*(i) for each $a$, the transpose of the $a$-th row of $\mathbf{Y}$, termed $\mathbf{y}^{(a)}$, satisfies*

$$[(\mathbf{D} - \alpha \mathbf{E})\mathbf{y}^{(a)}]_+ = \lambda \mathbf{y}^{(a)}, \ \forall a \in \{1, 2, \dots, m\}, \tag{5}$$

*(ii) Let $H$ be the stabilizer subgroup of $\mathbf{y}^{(1)}$, namely, $H = Stab\ \mathbf{y}^{(1)} \equiv \{h \in G | h\mathbf{y}^{(1)} = \mathbf{y}^{(1)}\}$. Then, $m = |G/H|$ and $\mathbf{Y}$ can be written as*

$$\mathbf{Y}^\top = \tfrac{1}{\sqrt{m}}[g_1 \mathbf{y}^{(1)} g_2 \mathbf{y}^{(1)} \cdots g_m \mathbf{y}^{(1)}], \tag{6}$$

*where $g_i$ are members of the $m$ distinct left cosets in $G/H$.*

In other words, when the symmetry group action is transitive, all the Lagrange multipliers are the same. Also the different rows of the $\mathbf{Y}$ matrix could be generated from a single row by the action of the group. A sketch of the proof is as follows (see Supplementary Material, Sec. A.3, for further

details). For part (i), we argue that a convex minimization problem with a symmetry always has a solution which respects the symmetry. Thus our search could be limited to the $G$-invariant elements of the convex cone, $\mathcal{CP}^G = \{\mathbf{Q} \in \mathcal{CP}^T \,|\, \mathbf{Q} = g\mathbf{Q}, \forall g \in G\}$, which happens to be a convex cone itself. We then introduce the Lagrange multipliers and define the Lagrangian for the problem on the invariant convex cone and show that it is enough to search over $\mathbf{\Lambda} = \lambda \mathbf{I}$. Part (ii) follows from optimality of $\mathbf{Q} = \mathbf{Y}^\top \mathbf{Y}$ implying optimality of $\bar{\mathbf{Q}} = \frac{1}{|G|} \sum_g g\mathbf{Q}$.

Eq. (5) is a non-linear eigenvalue equation that can have many solutions. Yet, if those solutions are related to each other by symmetry they can be found explicitly, as we show in the following subsections.

## 4.1 Solution for Inputs on the Ring with Cosine Similarity in the Continuum Limit

In this subsection, we consider the case where inputs, $\mathbf{x}_t$, lie on a one-dimensional manifold shaped as a ring centered on the origin:

$$\mathbf{x}_t = \left[\cos(\tfrac{2\pi t}{T}), \sin(\tfrac{2\pi t}{T})\right]^\top,$$

where $t \in \{1, 2, \ldots, T\}$. Then, we have $D_{tt'} = \cos\left[\tfrac{2\pi(t-t')}{T}\right]$ and Eq. (5) becomes

$$\left[\sum_{t'} \left(\cos\left[\tfrac{2\pi(t-t')}{T}\right] - \alpha\right) y_{at'}\right]_+ = \lambda y_{at}, \ \forall a \in \{1, 2, \ldots, m\}. \tag{7}$$

In the limit of large $T$, we can replace a discrete variable, $t$, by a continuous variable, $\theta$: $\tfrac{2\pi t}{T} \to \theta$, $D_{tt'} = \cos\left[\tfrac{2\pi(t-t')}{T}\right] \to \cos(\theta - \theta')$, $y_{at} \to Cu_\phi(\theta)$ and $\lambda \to T\mu$, leading to

$$\left[\frac{1}{2\pi} \int_0^{2\pi} \{\cos(\theta - \theta') - \alpha\} u_\phi(\theta') d\theta'\right]_+ = \mu u_\phi(\theta), \tag{8}$$

with $C$ adjusted so that $\int u_\phi(\theta)^2 d\mathfrak{m}(\phi) = 1$ for some measure $\mathfrak{m}$ in the space of $\phi$, which is a continuous variable labeling the output neurons. We will see that $\phi$ could naturally be chosen as an angle and the constraint becomes $\int_0^{2\pi} u_\phi(\theta)^2 d\phi = 1$.

Eq. (8) has appeared previously in the context of the ring attractor [21]. While our variables have a completely different neuroscience interpretation, we can still use their solution:

$$u_\phi(\theta) = A\left[\cos(\theta - \phi) - \cos(\psi)\right]_+ \tag{9}$$

whose support is the interval $[\phi - \psi, \phi + \psi]$.

Eq. (9) gives the receptive fields of a neuron, $\phi$, in terms of the azimuthal coordinate, $\theta$, shown in the bottom left panel of Fig. 1. The dependence of $\mu$ and $\psi$ on $\alpha$ is given parametrically (see Supplementary Material, Sec. A.4). So far, we have only shown that Eq. (9) satisfies the necessary optimality condition in the continuous limit of Eq. (8). In Sec. 6, we confirm numerically that the optimal solution for a finite number of neurons approximates Eq. (9), Fig. 2.

While we do not have a closed-form solution for NSM-0 on a ring, we show that the optimal solution also has localized receptive fields (see Supplementary Material, Sec. A.5).

## 4.2 Solution for Inputs on Higher-dimensional Compact Homogeneous Manifolds

Here, we consider two special cases of higher dimensional manifolds. The first example is the 2-sphere, $S^2 = SO(3)/SO(1)$. The second example is the rotation group, $SO(3)$, which is a three-dimensional manifold. It is possible to generalize this method to other compact homogeneous spaces for particular kernels.

We can think of a 2-sphere via its 3-dimensional embedding: $S^2 \equiv \{\mathbf{x} \in \mathbb{R}^3 | \|\mathbf{x}\| = 1\}$. For two points $\Omega, \Omega'$ on the 2-sphere let $\mathbf{D}(\Omega, \Omega') = \mathbf{x}(\Omega) \cdot \mathbf{x}(\Omega')$, where $\mathbf{x}(\Omega), \mathbf{x}(\Omega')$ are the corresponding unit vectors in the 3-dimensional embedding.

Remarkably, we can show that solutions satisfying the optimality conditions are of the form

$$u_{\Omega_0}(\Omega) = A\left[\mathbf{x}(\Omega_0) \cdot \mathbf{x}(\Omega) - \cos \psi\right]_+. \tag{10}$$

This means that the center of a receptive field on the sphere is at $\Omega_0$. The neuron is active while the angle between $\mathbf{x}(\Omega)$ and $\mathbf{x}(\Omega_0)$ is less than $\psi$. For the derivation of Eq. (10) and the self-consistency conditions, determining $\psi, \mu$ in terms of $\alpha$, see Supplementary Material, Sec. A.6.

In the case of the rotation group, for $g, g' \in SO(3)$ we adopt the $3 \times 3$ matrix representations $\mathbf{R}(g), \mathbf{R}(g')$ and consider $\frac{1}{3} \operatorname{Tr}\left(\mathbf{R}(g)\mathbf{R}(g')^\top\right)$ to be the similarity kernel. Once more, we index a receptive field solution by the rotation group element, $g_0$, where the response is maximum:

$$u_{g_0}(\Omega) = \frac{A}{2}\left[\operatorname{Tr}\left(\mathbf{R}(g_0)^\top \mathbf{R}(g)\right) - 2\cos\psi - 1\right]_+ \tag{11}$$

with $\psi, \mu$ being determined by $\alpha$ through self-consistency equations. This solution has support over $g \in SO(3)$, such that the rotation $gg_0^{-1}$ has a rotation angle less than $\psi$.

To summarize this section, we demonstrated, in the continuum limit, that the solutions to NSM objectives for data on symmetric manifolds possess localized receptive fields that tile these manifolds.

What is the nature of solutions as the datasets depart from highly symmetric cases? To address this question, consider data on a smooth compact Riemannian manifold with a smooth metric resulting in a continuous curvature tensor. Then the curvature tensor sets a local length scale over which the effect of curvature is felt. If a symmetry group acts transitively on the manifold, this length scale is constant all over the manifold. Even if such symmetries are absent, on a compact manifold, the curvature tensor components are bounded and there is a length scale, $L$, below which the manifold locally appears as flat space. Suppose the manifold is sampled well enough with many data points within each geodesic ball of length, $L$, and the parameters are chosen so that the localized receptive fields are narrower than $L$. Then, we could construct an asymptotic solution satisfying the optimality condition. Such asymptotic solution in the continuum limit and the effect of uneven sampling along the manifold will be analyzed elsewhere.

## 5 Online Optimization and Neural Networks

Here, we derive a biologically plausible neural network that optimizes NSM-1. To this end, we transform NSM-1 by, first, rewriting it in the Lagrangian form:

$$\min_{\forall t:\, \mathbf{y}_t \geq 0} \max_{\forall t:\, \mathbf{z}_t \geq 0} -\frac{1}{T}\sum_{t,t'}(\mathbf{x}_t \cdot \mathbf{x}_{t'} - \alpha)\mathbf{y}_t \cdot \mathbf{y}_{t'} + \sum_t \mathbf{z}_t \cdot \mathbf{z}_t(\mathbf{y}_t \cdot \mathbf{y}_t - \beta). \tag{12}$$

Here, unconventionally, the nonnegative Lagrange multipliers that impose the inequality constraints are factorized into inner products of two nonnegative vectors ($\mathbf{z}_t \cdot \mathbf{z}_t$). Second, we introduce auxiliary variables, $\mathbf{W}, \mathbf{b}, \mathbf{V}_t$ [10]:

$$\min_{\forall t:\, \mathbf{y}_t \geq 0} \max_{\forall t:\, \mathbf{z}_t \geq 0} \min_{\mathbf{W}} \max_{\mathbf{b}} \max_{\forall t:\, \mathbf{V}_t \geq 0} T \operatorname{Tr}(\mathbf{W}^\top \mathbf{W}) - T\|\mathbf{b}\|_2^2 + $$
$$+ \sum_t \left(-2\mathbf{x}_t \mathbf{W}^\top \mathbf{y}_t + 2\sqrt{\alpha}\mathbf{y}_t \cdot \mathbf{b} - \beta\|\mathbf{z}_t\|_2^2 + 2\mathbf{z}_t \mathbf{V}_t \mathbf{y}_t - \operatorname{Tr}(\mathbf{V}_t^\top \mathbf{V}_t)\right). \tag{13}$$

The equivalence of (13) to (12) can be seen by performing the $\mathbf{W}, \mathbf{b}$, and $\mathbf{V}_t$ optimizations explicitly and plugging the optimal values back. (13) suggests a two-step online algorithm (see Appendix A.8 for full derivation). For each input $\mathbf{x}_t$, in the first step, one solves for $\mathbf{y}_t, \mathbf{z}_t$ and $\mathbf{V}_t$, by projected gradient descent-ascent-descent,

$$\begin{bmatrix} \mathbf{y}_t \\ \mathbf{z}_t \\ \mathbf{V}_t \end{bmatrix} \longleftarrow \begin{bmatrix} \mathbf{y}_t + \gamma_y \left(\mathbf{W}\mathbf{x}_t - \mathbf{V}_t^\top \mathbf{z}_t - \sqrt{\alpha}\mathbf{b}\right) \\ \mathbf{z}_t + \gamma_z \left(-\beta\mathbf{z}_t + \mathbf{V}_t \mathbf{y}_t\right) \\ \mathbf{V}_t + \gamma_V \left(\mathbf{z}_t \mathbf{y}_t^\top - \mathbf{V}_t\right) \end{bmatrix}_+, \tag{14}$$

where $\gamma_{y,z,V}$ are step sizes. This iteration can be interpreted as the dynamics of a neural circuit (Fig. 1, Top right panel), where components of $\mathbf{y}_t$ are activities of excitatory neurons, $\mathbf{b}$ is a bias term, $\mathbf{z}_t$ – activities of inhibitory neurons, $\mathbf{W}$ is the feedforward connectivity matrix, and $\mathbf{V}_t$ is the synaptic weight matrix from excitatory to inhibitory neurons, which undergoes a fast time-scale anti-Hebbian plasticity. In the second step, $\mathbf{W}$ and $\mathbf{b}$ are updated by gradient descent-ascent:

$$\mathbf{W} \longleftarrow \mathbf{W} + \eta\left(\mathbf{y}_t \mathbf{x}_t^\top - \mathbf{W}\right), \qquad\qquad \mathbf{b} \longleftarrow \mathbf{b} + \eta\left(\sqrt{\alpha}\mathbf{y}_t - \mathbf{b}\right), \tag{15}$$

where $\mathbf{W}$ is going through a slow time-scale Hebbian plasticity and $\mathbf{b}$ through homeostatic plasticity. $\eta$ is a learning rate. Application of this algorithm to symmetric datasets is shown in Fig. 2 and Fig. 3.

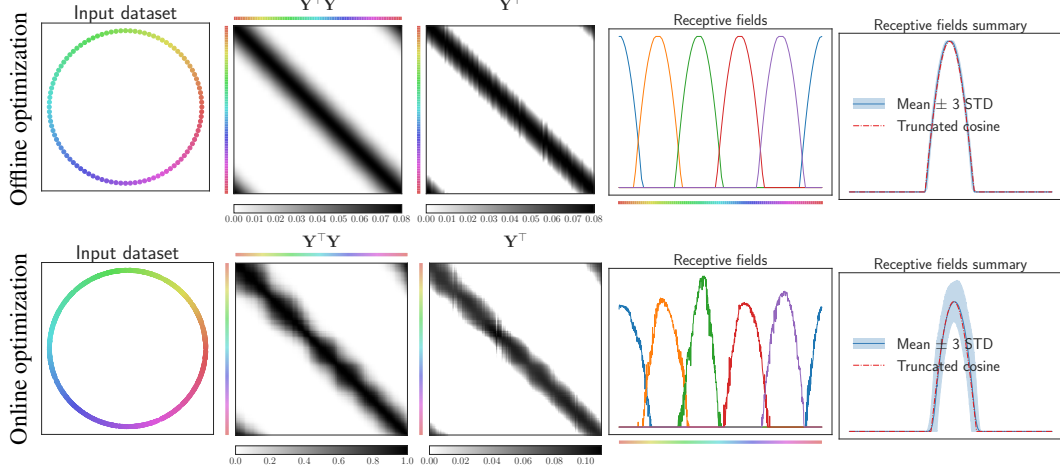

Figure 2: Solution of NSM-1 on a ring in 2D. **From left to right**, the input dataset $\mathbf{X}$, the output similarity, $\mathbf{Q}$, the output neural activity matrix $\mathbf{Y}$, a few localized receptive fields, and the aligned receptive fields. The receptive fields are truncated cosines translated along the ring.

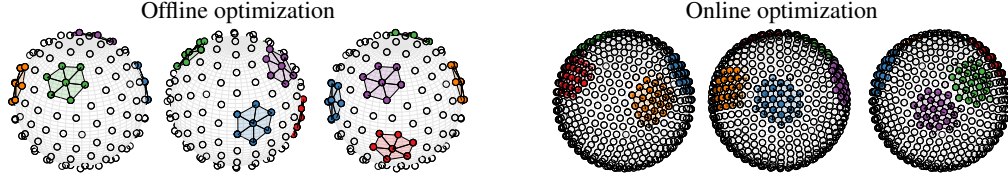

Figure 3: Solution of NSM-1 tiles the sphere with overlapping localized receptive fields (soft-clusters), providing an accurate and useful data representation. We show a few receptive fields in different colors over three different views of the sphere. An advantage of the online optimization is that it can handle arbitrarily large number of points.

## 6 Experimental Results

In this section, we verify our theoretical results by solving both offline and online optimization problems numerically. We confirm our theoretical predictions in Sec. 4 for symmetric manifolds and demonstrate that they hold for deviations from symmetry. Moreover, our algorithms yield manifold-tiling localized receptive fields on real-world data.

**Synthetic data.** Recall that for the input data lying on a ring, optimization without a rank constraint yields truncated cosine solutions, see Eq. (9). Here, we show numerically that fixed-rank optimization yields the same solutions, Fig. 2: the computed matrix $\mathbf{Y}^\top\mathbf{Y}$ is indeed circulant, all receptive fields are equivalent to each other, are well approximated by truncated cosine and tile the manifold with overlap. Similarly, for the input lying on a 2-sphere, we find numerically that localized solutions tile the manifold, Fig. 3.

For the offline optimization we used a Burer-Monteiro augmented Lagrangian method [23, 24]. Whereas, conventionally, the number of rows $m$ of $\mathbf{Y}$ is chosen to be $\beta T$ (observe that $\mathrm{diag}(\mathbf{Y}^\top\mathbf{Y}) \leq \beta\mathbf{I}$ implies that $\mathrm{Tr}(\mathbf{Y}^\top\mathbf{Y}) \leq \beta T$, making $\beta T$ an upper bound of the rank), we use the non-standard setting $m \gg \beta T$, as a small $m$ might create degeneracies (i.e., hard-clustering solutions).

Also, we empirically demonstrate that the nature of the solutions is robust to deviations from symmetry in manifold curvature and data point density. See Fig. 4 and its caption for details.

**Real-world data.** For normalized input data with every diagonal element $D_{tt} = \|\mathbf{x}_t\|_2^2$ above the threshold $\alpha$, the term $\alpha\,\mathrm{Tr}(\mathbf{EQ}) = \alpha\sum_{tt'}\mathbf{y}_t \cdot \mathbf{y}_{t'}$ in NSM-1 behaves as described in Sec. 2. For unnormalized inputs, it is preferable to control the sum of each row of $\mathbf{Q}$, i.e. $\sum_{t'}\mathbf{y}_t \cdot \mathbf{y}_{t'}$, with an individual $\alpha_t$, instead of the total sum.

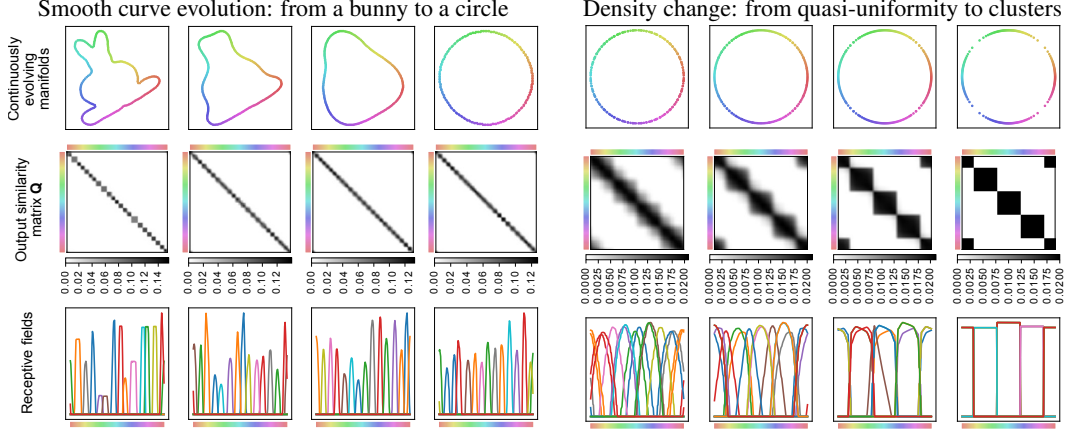

Figure 4: Robustness of the manifold-tiling solution to symmetry violations. **Left sequence:** Despite non-uniform curvature, the localized manifold-tiling nature of solutions is preserved for the wide range of datasets around the symmetric manifold. We start from a curve representing a bunny and evolve it using the classical mean curvature motion. **Right sequence:** Despite non-uniform point density, the localized manifold-tiling nature of solutions is preserved in the wide range of datasets around the symmetric manifold. For high density variation there is a smooth transition to the hard-clustering solution. The points are sampled from a mixture of von Mises distributions with means $0, \frac{\pi}{2}, \pi, \frac{3\pi}{2}$ and equal variance decreasing from left to right.

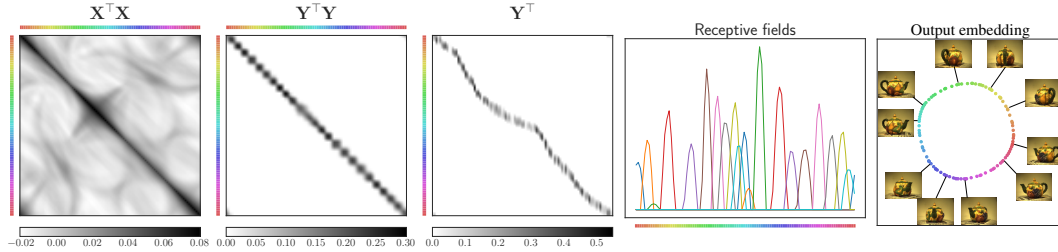

Figure 5: NSM-2 solution learns the manifold from a 100 images obtained by viewing a teapot from different angles. The obtained 1d manifold uncovers the change in orientation (better seen with zoom) by tiling it with overlapping localized receptive fields. The input size $n$ is 23028 ($76 \times 101$ pixels, 3 color channels). We build a 2d linear embedding (PCA) from the solution $\mathbf{Y}$.

Additionally, enforcing $\sum_t \|\mathbf{y}_t\|_2^2 \leq \beta T$ is in many cases empirically equivalent to enforcing $\|\mathbf{y}_t\|_2^2 \leq \beta$ but makes the optimization easier. We thus obtain the objective function

$$\min_{\substack{\forall t: \, \mathbf{y}_t \geq 0, \\ \sum_t \|\mathbf{y}_t\|_2^2 \leq \beta T}} \quad -\sum_{t,t'} (\mathbf{x}_t \cdot \mathbf{x}_{t'} - \alpha_t) \mathbf{y}_t \cdot \mathbf{y}_{t'}, \tag{16}$$

which, for some choice of $\alpha_t$, is equivalent to (here, $\mathbf{1} \in \mathbb{R}^T$ is a column vector of ones)

$$\min_{\mathbf{Y} \geq 0} -\operatorname{Tr}(\mathbf{X}^\top \mathbf{X} \mathbf{Y}^\top \mathbf{Y}) \quad \text{s.t.} \quad \mathbf{Y}^\top \mathbf{Y} \mathbf{1} = \mathbf{1}, \; \operatorname{Tr}(\mathbf{Y}^\top \mathbf{Y}) \leq \beta T, \tag{NSM-2}$$

For highly symmetric datasets without constraints on rank, NSM-2 has the same solutions as NSM-1 (see Supplementary Material, Sec. A.7). Relaxations of this optimization problem have been the subject of extensive research to solve clustering and manifold learning problems [25, 26, 27, 28]. A biologically plausible neural network solving this problem was proposed in [12]. For the optimization of NSM-2 we use an augmented Lagrangian method [23, 24, 28, 29].

We have extensively applied NSM-2 to datasets previously analyzed in the context of manifold learning [28, 30] (see Supplementary Material, Sec. B). Here, we include just two representative examples, figs. 5 and 6, showing the emergence of localized receptive fields in a high-dimensional space. Despite the lack of symmetry and ensuing loss of regularity, we obtain neurons whose receptive

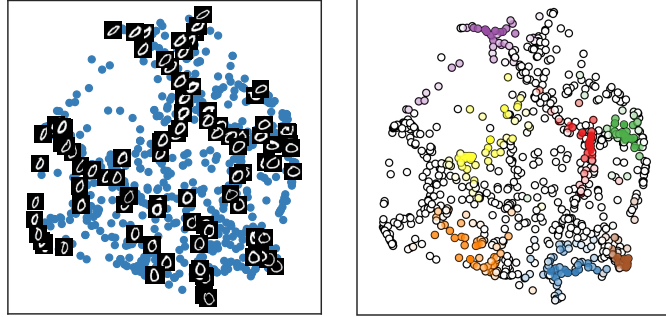

Figure 6: NSM-2 solution learns the manifold of MNIST digit 0 images by tiling the dataset with overlapping localized receptive fields. Input size is $n = 28 \times 28 = 784$. **Left:** Two-dimensional linear embedding (PCA) of $\mathbf{Y}$. The data gets organized according to different visual characteristics of the hand-written digit (e.g., orientation and elongation). **Right:** A few receptive fields in different colors over the low-dimensional embedding.

fields, taken together, tile the entire data cloud. Such tiling solutions indicate robustness of the method to imperfections in the dataset and further corroborate the theoretical results derived in this paper.

# 7 Discussion

In this work, we show that objective functions approximately preserving similarity, along with nonnegativity constraint on the outputs, learn data manifolds. Neural networks implementing NSM algorithms use only biologically plausible local (Hebbian or anti-Hebbian) synaptic learning rules.

These results add to the versatility of NSM networks previously shown to cluster data, learn sparse dictionaries and blindly separate sources [11, 18, 16], depending on the nature of input data. This illustrates how a universal neural circuit in the brain can implement various learning tasks [11].

Our algorithms, starting from a linear kernel, $\mathbf{D}$, generate an output kernel, $\mathbf{Q}$, restricted to the sample space. Whereas the associations between kernels and neural networks was known [31], previously proposed networks used random synaptic weights with no learning. In our algorithms, the weights are learned from the input data to optimize the objective. Therefore, our algorithms learn data-dependent kernels adaptively.

In addition to modeling biological neural computation, our algorithms may also serve as general-purpose mechanisms for generating representations of manifolds adaptively. Unlike most existing manifold learning algorithms [32, 33, 34, 35, 36, 37], ours can operate naturally in the online setting. Also, unlike most existing algorithms, ours do not output low-dimensional vectors of embedding variables but rather high-dimensional vectors of assignment indices to centroids tiling the manifold, similar to radial basis function networks [38]. This tiling approach is also essentially different from setting up charts [39, 40], which essentially end up modeling local tangent spaces. The advantage of our high-dimensional representation becomes obvious if the output representation is used not for visualization but for further computation, e.g., linear classification [41].

**Acknowledgments**

We are grateful to Yanis Bahroun, Johannes Friedrich, Victor Minden, Eftychios Pnevmatikakis, and the other members of the Flatiron Neuroscience group for discussion and comments on an earlier version of this manuscript. We thank Sanjeev Arora, Afonso Bandeira, Moses Charikar, Jeff Cheeger, Surya Ganguli, Dustin Mixon, Marc'Aurelio Ranzato, and Soledad Villar for helpful discussions.

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
