[Supplementary Material]

# Manifold-tiling Localized Receptive Fields are Optimal in Similarity-preserving Neural Networks

## Supplementary Material

## A Details of Derivations

### A.1 Relation between NSM-0 and NSM-1

We will consider the optimization under an additional power constraint: $\operatorname{Tr} \mathbf{Y}^\top \mathbf{Y} \leq k$ as in [9, 11]. To arrive at a simpler objective function, let us first scale the current objective function by an overall of factor of $\frac{\beta}{kT}$, a procedure that does not affect the optimal $\mathbf{Y}$:

$$\underset{\substack{\mathbf{Y} \geq 0 \\ \operatorname{Tr} \mathbf{Y}^\top \mathbf{Y} \leq k}}{\operatorname{argmin}} \frac{\beta}{2kT} \|\mathbf{X}^\top \mathbf{X} - \alpha \mathbf{E} - \mathbf{Y}^\top \mathbf{Y}\|_F^2 = \underset{\substack{\mathbf{Y} \geq 0 \\ \operatorname{Tr} \mathbf{Y}^\top \mathbf{Y} \leq k}}{\operatorname{argmin}} -\frac{\beta}{kT} \operatorname{Tr}((\mathbf{X}^\top \mathbf{X} - \alpha \mathbf{E}) \mathbf{Y}^\top \mathbf{Y}) + \frac{\beta}{2kT} \|\mathbf{Y}^\top \mathbf{Y}\|_F^2.$$
(17)

Now we rescale the variables: $\tilde{\mathbf{Y}} = \sqrt{\frac{\beta T}{k}} \mathbf{Y}$. If we hold $\beta$ fixed and let $k \to 0$, the optimal $\tilde{\mathbf{Y}}$ is given by

$$\lim_{k \to 0} \underset{\substack{\tilde{\mathbf{Y}} \geq 0 \\ \operatorname{Tr} \tilde{\mathbf{Y}}^\top \tilde{\mathbf{Y}} \leq \beta T}}{\operatorname{argmin}} -\operatorname{Tr}((\mathbf{X}^\top \mathbf{X} - \alpha \mathbf{E}) \tilde{\mathbf{Y}}^\top \tilde{\mathbf{Y}}) + \frac{kT}{2\beta} \|\tilde{\mathbf{Y}}^\top \tilde{\mathbf{Y}}\|_F^2 = \underset{\substack{\tilde{\mathbf{Y}} \geq 0 \\ \operatorname{Tr} \tilde{\mathbf{Y}}^\top \tilde{\mathbf{Y}} \leq \beta T}}{\operatorname{argmin}} -\operatorname{Tr}((\mathbf{X}^\top \mathbf{X} - \alpha \mathbf{E}) \tilde{\mathbf{Y}}^\top \tilde{\mathbf{Y}},$$
(18)

since, in that limit, we can ignore the second term and just minimize the quadratic function of $\tilde{\mathbf{Y}}$.

We will work with the quadratic objective function discussed above. Switching back to $\mathbf{Y}$ from $\tilde{\mathbf{Y}}$ and applying a constraint on each diagonal element of $\mathbf{Y}^\top \mathbf{Y}$ rather than to $\operatorname{Tr} \mathbf{Y}^\top \mathbf{Y}$, we have:

$$\min_{\substack{\mathbf{Y} \geq 0 \\ \operatorname{diag}(\mathbf{Y}^\top \mathbf{Y}) \leq \beta \mathbf{I}}} -\operatorname{Tr}((\mathbf{D} - \alpha \mathbf{E}) \mathbf{Y}^\top \mathbf{Y}) = \min_{\substack{\forall t \in \{1,2,\ldots,T\}, \, \mathbf{y}_t \geq 0 \\ \|\mathbf{y}_t\|_2^2 \leq \beta}} -\sum_{t,t'} (\mathbf{x}_t \cdot \mathbf{x}_{t'} - \alpha) \mathbf{y}_t \cdot \mathbf{y}_{t'}.$$
(19)

### A.2 Proof of Proposition 1

**Proposition 1.** *The optimal solution of Problem (NSM-1) satisfies*

$$[(\mathbf{D} - \alpha \mathbf{E}) \mathbf{y}^{(a)}]_+ = \mathbf{\Lambda} \mathbf{y}^{(a)},$$
(20)

*where $\mathbf{y}^{(a)}$ refers to the transpose of the $a$-th row of $\mathbf{Y}$ as a column vector and $\mathbf{\Lambda} = \operatorname{diag}(\lambda_1, \ldots, \lambda_T)$ is a nonnegative diagonal matrix.*

*Proof.* We introduce Lagrange multipliers $\mathbf{\Lambda} = \operatorname{diag}(\lambda_1, \ldots, \lambda_T)$ for the constraint, $\operatorname{diag}(\mathbf{Y}^\top \mathbf{Y}) \leq \beta \mathbf{I}$ and the nonnegative Lagrange multiplier matrix $\mathbf{Z} \in R^{m \times T}$, $\mathbf{Z} \geq 0$, for the constraint $\mathbf{Y} \geq 0$:

$$\min_{\substack{\mathbf{Y} \geq 0 \\ \operatorname{diag}(\mathbf{Y}^\top \mathbf{Y}) \leq \beta \mathbf{I}}} -\operatorname{Tr}((\mathbf{D} - \alpha \mathbf{E}) \mathbf{Y}^\top \mathbf{Y})$$

$$= \min_{\mathbf{Y} \geq 0} \max_{\mathbf{\Lambda} \geq 0} \operatorname{Tr} \left( -(\mathbf{D} - \alpha \mathbf{E}) \mathbf{Y}^\top \mathbf{Y} + \mathbf{\Lambda} (\mathbf{Y}^\top \mathbf{Y} - \beta \mathbf{I}) \right)$$

$$= \min_{\mathbf{Y}} \max_{\mathbf{\Lambda} \geq 0, \mathbf{Z} \geq 0} \operatorname{Tr} \left( -(\mathbf{D} - \alpha \mathbf{E}) \mathbf{Y}^\top \mathbf{Y} + \mathbf{\Lambda} (\mathbf{Y}^\top \mathbf{Y} - \beta \mathbf{I}) \right) - \operatorname{Tr}(\mathbf{Z}^\top \mathbf{Y}).$$
(21)

By varying $\mathbf{Y} \in \mathbb{R}^{m \times T}$ we derive a necessary optimality condition:

$$(\mathbf{D} - \alpha \mathbf{E} - \mathbf{\Lambda}) \mathbf{Y}^\top = -\mathbf{Z}^\top / 2.$$
(22)

Since $\mathbf{Z}$ entries, $z_{at}$, are nonnegative, Eq. (22) implies that the terms of the matrix $(\mathbf{D} - \alpha \mathbf{E} - \mathbf{\Lambda}) \mathbf{Y}^\top$ are either zero or negative. More over, if $y_{at} > 0$ in the final solution, then the corresponding Lagrange multiplier entry of the $\mathbf{Z}$ matrix, $z_{at}$, is zero. We will use this fact to simplify the equations.

To analyze Eq. (22) further, it is convenient to designate, analogically to $\mathbf{y}^{(a)}$, the transpose of the $a$-th row of $\mathbf{Z}$ as a column vector, $\mathbf{z}^{(a)}$. Each $\mathbf{y}^{(a)}$ must now satisfy:

$$(\mathbf{D} - \alpha\mathbf{E} - \boldsymbol{\Lambda})\mathbf{y}^{(a)} = -\mathbf{z}^{(a)}/2. \tag{23}$$

Now consider the two possibilities:

- If $\lambda_t y_{at} > 0$, then $y_{at} > 0$. Optimality conditions requires that $z_{at}$ is zero, implying:

$$\left((\mathbf{D} - \alpha\mathbf{E})\mathbf{y}^{(a)}\right)_t = \lambda_t y_{at}. \tag{24}$$

  Since the RHS is positive, the LHS has to be positive. We can, therefore, replace the LHS by $[((\mathbf{D} - \alpha\mathbf{E})\mathbf{y}^{(a)})_t]_+$, giving us

$$[((\mathbf{D} - \alpha\mathbf{E})\mathbf{y}^{(a)})_t]_+ = \lambda_t y_{at}. \tag{25}$$

- If $\lambda_t y_{at} = 0$, we have

$$\left((\mathbf{D} - \alpha\mathbf{E})\mathbf{y}^{(a)}\right)_t \leq 0. \tag{26}$$

  In this case, we trivially get

$$[((\mathbf{D} - \alpha\mathbf{E})\mathbf{y}^{(a)})_t]_+ = 0 = \lambda_t y_{at}. \tag{27}$$

Thus $[((\mathbf{D} - \alpha\mathbf{E})\mathbf{y}^{(a)})_t]_+ = \lambda_t y_{at}$ for all $a, t$. Noting that $(\boldsymbol{\Lambda}\mathbf{y}^{(a)})_t = \lambda_t y_{at}$,

$$[(\mathbf{D} - \alpha\mathbf{E})\mathbf{y}^{(a)}]_+ = \boldsymbol{\Lambda}\mathbf{y}^{(a)}. \tag{28}$$

$\square$

The Lagrange multipliers $\lambda_1, \ldots, \lambda_T$ must be adjusted so that $\sum_a y_{at}^2 \leq \beta$, for each $t$, with $\lambda_t = 0$, when the inequality is strict, i.e., $\sum_a y_{at}^2 < \beta$.

### A.3 Proof of Theorem 1

Invariance under group action could be represented concisely with the help of matrices $\{\mathbf{R}(g) \,|\, g \in G\}$, which form the natural representation of the group $G$: $\mathbf{R}(g)_{tt'} = \delta_{g(t)t'}$, $\delta_{mn}$ being the Kronecker delta. The condition of symmetry for the problem is then stated very simply: $\mathbf{D} = \mathbf{R}(g)\mathbf{D}\mathbf{R}(g)^\top$.

**Lemma 1.** *To optimize Problem (NSM-1a), it is possible to restrict the search to the $G$-invariant convex cone $\mathcal{CP}^G = \{\mathbf{Q} \in \mathcal{CP}^T \,|\, \mathbf{Q} = \mathbf{R}(g)\mathbf{Q}\mathbf{R}(g)^\top, \forall g \in G\}$.*

*Proof.* Thanks to convexity of Problem (NSM-1a), there exists one optimal solution $\mathbf{Q}_0$ which is invariant under $G$, namely $\mathbf{R}(g)\mathbf{Q}_0\mathbf{R}(g)^\top = \mathbf{Q}_0$ for all $g \in G$ [42].[2] Hence, it is enough to search for solutions that are invariant under the group $G$. Since we search for solutions that are invariant, we restrict ourselves to the convex cone of $G$-invariant $T \times T$ completely positive matrices: $\mathcal{CP}^G = \{\mathbf{Q} \in \mathcal{CP}^T \,|\, \mathbf{Q} = \mathbf{R}(g)\mathbf{Q}\mathbf{R}(g)^\top, \forall g \in G\}$. $\square$

Before getting to our main theorem, we make a few comments on finite group action on vectors and the number of distinct vectors generated. Let $\mathbf{y} \in R^T$. The action of $g \in G$ on $\mathbf{y}$ is given by $g\mathbf{y} = \mathbf{R}(g)\mathbf{y}$. The orbit of $\mathbf{y}$ under the action of $G$, $G\mathbf{y} = \{g\mathbf{y}|g \in G\}$, is a set on which $G$ acts transitively. The cardinality of the set $G\mathbf{y}$, in other words, the number of distinct vectors produced by action of $G$ on $\mathbf{y}$, is given by the index of the stabilizer subgroup of $\mathbf{y}$, $Stab\, \mathbf{y} = \{h \in G|h\mathbf{y} = \mathbf{y}\}$, in the group $G$. The index of a subgroup $H$ in $G$, $[G:H]$ is the number of left cosets in the coset space $G/H$ [43]. Summarizing,

$$|G\mathbf{y}| = [G : Stab\, \mathbf{y}] = |G/Stab\, \mathbf{y}| = |G|/|Stab\, \mathbf{y}|.$$

Sometimes, this observation is called the orbit-stabilizer formula or theorem. Also, the distinct entries of $G\mathbf{y}$ are created by the action of members of distinct left cosets of $Stab\, \mathbf{y}$. In our results, we will use the cardinality of $G/H$, $|G/H|$ rather than the $[G:H]$ to indicate the index.

**Theorem 1.** *If the action of the group $G$ is transitive, that is, for any pair $t, t' \in \{1, 2, \ldots, T\}$ there is a $g \in G$ so that $t' = g(t)$, then there is at least one optimal solution of Problem (NSM-1a) with $\mathbf{Q} = \mathbf{Y}^\top \mathbf{Y}, \mathbf{Y} \in \mathbb{R}^{m \times T}$ and $\mathbf{Y} \geq 0$, such that*

*(i) for each $a$, the transpose of the $a$-th row of $\mathbf{Y}$, termed $\mathbf{y}^{(a)}$, satisfies*

$$[(\mathbf{D} - \alpha \mathbf{E})\mathbf{y}^{(a)}]_+ = \lambda \mathbf{y}^{(a)}, \forall a \in \{1, 2, \ldots, m\}, \tag{29}$$

*(ii) Let $H$ be the stabilizer subgroup of $\mathbf{y}^{(1)}$, namely, $H = Stab\, \mathbf{y}^{(1)} \equiv \{h \in G | h\mathbf{y}^{(1)} = \mathbf{y}^{(1)}\}$. Then, $m = |G/H|$ and $\mathbf{Y}$ can be written as*

$$\mathbf{Y}^\top = \tfrac{1}{\sqrt{m}}[g_1 \mathbf{y}^{(1)} g_2 \mathbf{y}^{(1)} \ldots g_m \mathbf{y}^{(1)}], \tag{30}$$

*where $g_i$ are members of the $m$ distinct left cosets in $G/H$.*

*Proof.* Part (i): Let us go back to the Lagrangian for Problem (NSM-1a):

$$\underset{\substack{\mathbf{Q} \in \mathcal{CP}^G \\ \mathrm{diag}(\mathbf{Q}) \leq \beta \mathbf{I}}}{\mathrm{argmin}} -\mathrm{Tr}((\mathbf{D} - \alpha \mathbf{E})\mathbf{Q}) = \underset{\mathbf{Q} \in \mathcal{CP}^G}{\mathrm{argmin}} \underset{\substack{\mathbf{\Lambda} = \mathrm{diag}(\lambda_1, \ldots, \lambda_T) \\ \mathbf{\Lambda} \geq 0}}{\mathrm{argmax}} -\mathrm{Tr}((\mathbf{D} - \alpha \mathbf{E})\mathbf{Q}) + \mathrm{Tr}(\mathbf{\Lambda}(\mathbf{Q} - \beta \mathbf{I})).$$

$$\tag{31}$$

Following, Lemma 1, as we are only considering $\mathbf{Q}$ belonging to the $G$-invariant convex cone $\mathcal{CP}^G$, we can replace $\mathbf{Q}$ by its orbit average $\frac{1}{|G|}\sum_{g \in G} \mathbf{R}(g)\mathbf{Q}\mathbf{R}(g)^\top$ in the expression in Eq. (31). Looking at the $\mathrm{Tr}(\mathbf{\Lambda}(\mathbf{Q} - \beta\mathbf{I}))$ term with this replacement, it is clear that in the Lagrangian, we could replace $\mathbf{\Lambda}$ by $\frac{1}{|G|}\sum_{g \in G} \mathbf{R}(g)\mathbf{\Lambda}\mathbf{R}(g)^\top$. So we can restrict ourselves to Lagrange multipliers that are $G$-invariant. In other words, we can assume $\mathbf{\Lambda} = \mathbf{R}(g)\mathbf{\Lambda}\mathbf{R}(g)^\top$.

Since the group action is transitive, the invariance implies that the diagonal elements of $\mathbf{\Lambda}$ are the same, giving us $\mathbf{\Lambda} = \lambda \mathbf{I}$. Now, using Proposition 1 and $\mathbf{Q} \in \mathcal{CP}^G$, the optimality condition becomes

$$[(\mathbf{D} - \alpha \mathbf{E})\mathbf{y}^{(a)}]_+ = \lambda \mathbf{y}^{(a)}. \tag{32}$$

Part (ii): This part is straightforward if tedious. Imagine we have a $G$-invariant $\mathbf{Q} = \mathbf{Y}_0^T \mathbf{Y}_0 = \sum_{c=1}^{m_0} \mathbf{y}_0^{(c)} \mathbf{y}_0^{(c)\top}$ which is an optimal solution. The nonnegative vectors $\mathbf{y}_0^{(c)\top}$ form the rows of the matrix $\mathbf{Y}_0 \in R^{m_0 \times T}$, and could be assumed to be nonzero vectors without loss of generality.

Since $\mathbf{Q}$ is $G$-invariant we have,

$$\mathbf{Q} = \tfrac{1}{|G|}\sum_{g \in G} \mathbf{R}(g)\mathbf{Q}\mathbf{R}(g)^\top = \tfrac{1}{|G|}\sum_{g \in G} \mathbf{R}(g)\mathbf{Y}_0^T \mathbf{Y}_0 \mathbf{R}(g)^\top. \tag{33}$$

Rewrite the last matrix $\frac{1}{|G|}\sum_{g \in G} \mathbf{R}(g)\mathbf{Y}_0^T \mathbf{Y}_0 \mathbf{R}(g)^\top$ as $\sum_{c=1}^{m_0} \frac{1}{|G|}\sum_{g \in G} \mathbf{R}(g)\mathbf{y}_0^{(c)}(\mathbf{R}(g)\mathbf{y}_0^{(c)})^\top$. Let $\tilde{\mathbf{Q}}_c \equiv \frac{\mathrm{Tr}(\mathbf{Q})}{|G|\|\mathbf{y}_0^{(c)}\|_2^2}\sum_{g \in G} \mathbf{R}(g)\mathbf{y}_0^{(c)}(\mathbf{R}(g)\mathbf{y}_0^{(c)})^\top$.

Then $\mathrm{Tr}(\mathbf{Q}_c) = \mathrm{Tr}(\mathbf{Q})$ indicating the diagonal elements of $\mathbf{Q}_c$ are the same as that of $\mathbf{Q}$. Also $\mathbf{Q}$ is a convex combination of $\mathbf{Q}_c$'s.

$$\mathbf{Q} = \sum_{c=1}^{m_0} \tfrac{1}{|G|}\sum_{g \in G} \mathbf{R}(g)\mathbf{y}_0^{(c)}(\mathbf{R}(g)\mathbf{y}_0^{(c)})^\top = \sum_{c=1}^{m_0} \rho_c \mathbf{Q}_c, \tag{34}$$

where $\rho_c = \frac{\|\mathbf{y}_0^{(c)}\|_2^2}{\mathrm{Tr}(\mathbf{Q})} = \frac{\|\mathbf{y}_0^{(c)}\|_2^2}{\sum_{c'=1}^{m_0} \|\mathbf{y}_0^{(c')}\|_2^2}$. Note that $\sum_c \rho_c = 1$. We will show that one of the $\mathbf{Q}_c$ is as good a solution as $\mathbf{Q}$.

Let $c_0 = \mathrm{argmax}_c \mathrm{Tr}((\mathbf{D} - \alpha \mathbf{E})\mathbf{Q}_c)$. Since each $\mathbf{Q}_c$ satisfies the same constraints as $\mathbf{Q}$ and $-\mathrm{Tr}((\mathbf{D} - \alpha \mathbf{E})\mathbf{Q}) = -\sum_c \rho_c \mathrm{Tr}((\mathbf{D} - \alpha \mathbf{E})\mathbf{Q}_c) \geq -\mathrm{Tr}((\mathbf{D} - \alpha \mathbf{E})\mathbf{Q}_{c_0})$, $\mathbf{Q}_{c_0}$ is an optimal solution.

Now let $\mathbf{y}^{(1)} \equiv \frac{\sqrt{\mathrm{Tr}(\mathbf{Q})}}{\|\mathbf{y}_0^{(c_0)}\|_2}\mathbf{y}_0^{(c_0)}$. Construct $\mathbf{Y}$ such that

$$\mathbf{Y}^\top = \tfrac{1}{\sqrt{|G/H|}}[g_1 \mathbf{y}^{(1)} g_2 \mathbf{y}^{(1)} \ldots g_{|G/H|} \mathbf{y}^{(1)}]$$

with $g_i$ belonging to distinct cosets in $G/H$, where $H$ is the stabilizer subgroup of $\mathbf{y}_1$.

With that, we have

$$\mathbf{Y}^\top \mathbf{Y} = \tfrac{1}{|G/H|} \sum_{g \in \{g_1, g_2, \ldots, g_{|G/H|}\}} \mathbf{R}(g)\mathbf{y}^{(1)}(\mathbf{R}(g)\mathbf{y}^{(1)})^\top = \tfrac{1}{|G|} \sum_{g \in G} \mathbf{R}(g)\mathbf{y}^{(1)}(\mathbf{R}(g)\mathbf{y}^{(1)})^\top = \mathbf{Q}_{c_0}, \tag{35}$$

which is an optimal solution. $\qquad\square$

### A.4 Details of Self-consistency Condition in the Ring Solution

In this section we provide additional details about the derivations presented in Sec. 4.1. Let us focus on the solution for $\phi = 0$,

$$\left[ \tfrac{1}{2\pi} \int_0^{2\pi} \big\{ \cos(\theta - \theta') - \alpha \big\} u_0(\theta') d\theta' \right]_+ = \mu u_0(\theta). \tag{36}$$

Using $\cos(\theta - \theta') = \cos\theta \cos\theta' + \sin\theta \sin\theta'$ and our ansatz for $u_0(\theta')$ being even in $\theta'$, we get

$$\left[ \tfrac{\cos\theta}{2\pi} \int_0^{2\pi} u_0(\theta') \cos\theta' d\theta' - \tfrac{\alpha}{2\pi} \int_0^{2\pi} u_0(\theta') d\theta' \right]_+ = \mu u_0(\theta). \tag{37}$$

The LHS is already in the form $A[\cos\theta - \cos\psi]_+$. Hence,

$$\cos\psi = \frac{\alpha \int_0^{2\pi} u_0(\theta') d\theta'}{\int_0^{2\pi} u_0(\theta') \cos\theta' d\theta'} = \frac{\alpha \int_0^{\psi} (\cos\theta' - \cos\psi) d\theta'}{\int_0^{\psi} (\cos\theta' - \cos\psi) \cos\theta' d\theta'}. \tag{38}$$

We now compute $\mu$ in terms of $\alpha, \psi$ by setting $\theta = 0$ in Eq. (37). Remembering that $u_0(\theta')$ is proportional to $[\cos\theta' - \cos\psi]_+$

$$\frac{1}{\pi} \int_0^{\psi} (\cos\theta' - \cos\psi) \cos\theta' d\theta' - \frac{\alpha}{\pi} \int_0^{\psi} (\cos\theta' - \cos\psi) d\theta' = \mu(1 - \cos\psi). \tag{39}$$

Using Eq. (38), this can be simplified to

$$\mu = \tfrac{1}{\pi} \int_0^{\psi} (\cos\theta' - \cos\psi) \cos\theta' d\theta'. \tag{40}$$

Doing the integrals, we get the dependence of $\mu$ and $\psi$ on $\alpha$, given parametrically by

$$\alpha = \frac{\cos\psi(2\psi - \sin 2\psi)}{4(\sin\psi - \psi\cos\psi)}, \tag{41} \qquad\qquad \mu = \tfrac{1}{4\pi}(2\psi - \sin 2\psi). \tag{42}$$

### A.5 Optimality for NSM-0 and the Solution on the Ring

In this section we provide additional details about the derivations presented in Sec. 4.1. If we go back to a nonnegative version of similarity-preserving, we could rewrite the problem as:

$$\operatorname*{argmin}_{\substack{\mathbf{Q} \in \mathcal{CP} \\ \operatorname{Tr}(\mathbf{Q}) \leq k}} \tfrac{1}{2} \operatorname{Tr}\big((\mathbf{D} - \alpha\mathbf{E} - \mathbf{Q})^2\big). \tag{NSM-0a}$$

We have chosen to use an inequality $\operatorname{Tr}(\mathbf{Q}) = \operatorname{Tr}(\mathbf{Y}^\top\mathbf{Y}) \leq k$ rather than an equality to simplify the discussion. Note that these optimization problems minimize a convex function, in fact a quadratic one in $\mathbf{Q}$, in a convex region. Many of the arguments we have made so far about invariant problems apply to these problems as well. If we derive the optimality condition following our earlier arguments, we get

$$[(\mathbf{D} - \alpha\mathbf{E} - \mathbf{Q})\mathbf{y}^{(a)}]_+ = \lambda\mathbf{y}^{(a)}. \tag{43}$$

The choice of $\lambda$ decides the constraint on $\operatorname{Tr}(\mathbf{Y}^\top\mathbf{Y})$. If we set $\lambda$ to be near zero, we get $\mathbf{Q}$ as close to $\mathbf{D} - \alpha\mathbf{E}$ as possible for the elements where $D - \alpha E$ is nonnegative. As we let $\lambda$ grow bigger, $\mathbf{Y}$

becomes smaller. As $\lambda$ gets closer to its maximum possible value, we have $\mathbf{D} - \alpha\mathbf{E} \gg \mathbf{Y}^\top\mathbf{Y} = \mathbf{Q}$. In that limit,

$$[(\mathbf{D} - \alpha\mathbf{E})\mathbf{y}^{(a)}]_+ = \lambda_m\mathbf{y}^{(a)}, \tag{44}$$

which is the nonnegative eigenvector problem we saw for NSM-1 when it has a symmetry group acting transitively. The Lagrange multiplier $\lambda$ thus takes values between zero and $\lambda_m$.

In NSM-0, $\alpha\mathbf{E} + \mathbf{Q}$ provides the inhibitory interaction to make the localized bump solutions. The non-triviality of the analysis comes from having to solve $\mathbf{y}^{(a)}$'s to determine $\mathbf{Q}$ and *vice versa*. That self-consistency problem is non-trivial, in general.

For the ring model, thanks to the symmetries, we have $\mathbf{Q}$ to be a circulant matrix. In this case, we can solve the self-consistency problem numerically quite easily. As shown in Fig. 7, increasing $\lambda$ reduces

Figure 7: The receptive fields from NSM-0 for two different values of $\lambda$. We use $T = 628$, but plot the result against $\theta = 2\pi t/T - \pi$.

the size of the response but also alters the tuning curve. When $\lambda \approx \lambda_m$, $\mathbf{y}^{(a)}$ is approximately given by truncated cosines.

## A.6 Details of NSM-1 Solutions for $S^2$ and for $SO(3)$

In this section we provide additional details about the derivations presented in Sec. 4.2. For $S^2$, we start from the equation

$$\frac{1}{4\pi}\left[\int_{S^2}\{\mathbf{x}(\Omega)\cdot\mathbf{x}(\Omega') - \alpha)\}u_\sigma(\Omega')d^2\Omega'\right]_+ = \mu u_\sigma(\Omega). \tag{45}$$

This equation invariant under 3-dimensional rotations around the origin of the sphere. For $g \in SO(3)$, let the action of $g$ take $\Omega$ to $g\Omega$, with $\mathbf{x}(g\Omega) = \mathbf{R}(g)\mathbf{x}(\Omega)$. $\mathbf{R}(g)$ is the $3 \times 3$ orthogonal matrix representation of the rotation. If $u_\sigma(\Omega)$ is a solution, so is $u_\sigma(g\Omega)$. We will use this feature to create a family of solutions by action of rotation group on a single solution.

Since $\int_{S^2} \mathbf{x}(\Omega) \cdot \mathbf{x}(\Omega') u_\sigma(\Omega') d^2\Omega' = \mathbf{x}(\Omega) \cdot \left[ \int_{S^2} \mathbf{x}(\Omega') u_\sigma(\Omega') d^2\Omega' \right]$, we will compute the vector $\int_{S^2} \mathbf{x}(\Omega') u_\sigma(\Omega') d^2\Omega'$. One way to describe the point $\Omega$ on the sphere is by using polar coordinates $(\theta, \phi)$, with $\mathbf{x}(\Omega) = (\sin\theta\cos\phi, \sin\theta\sin\phi, \cos\theta)$.

Imagine $u_\sigma(\Omega) = U(\theta)$, namely, it is independent of $\phi$. In that case,

$$\int_{S^2} \mathbf{x}(\Omega') u_\sigma(\Omega') d^2\Omega' = \left( 0, 0, 2\pi \int_0^\pi d\theta' \sin\theta' \cos\theta' U(\theta') \right), \tag{46}$$

implying that $\int_{S^2} \mathbf{x}(\Omega) \cdot \mathbf{x}(\Omega') u_\sigma(\Omega') d^2\Omega' \propto x_3(\Omega) = \cos(\theta)$. Thus,

$$\int_{S^2} \left\{ \mathbf{x}(\Omega) \cdot \mathbf{x}(\Omega') - \alpha \right\} u_\sigma(\Omega') d^2\Omega' = a\cos\theta - b, \tag{47}$$

with $a = 2\pi \int_0^\pi d\theta' \sin\theta' \cos\theta' U(\theta')$ and $b = 2\alpha\pi \int_0^\pi d\theta' \sin\theta' U(\theta')$. Defining $A = a/\mu, B = b/\mu$, and using Eq. (45), we have

$$U(\theta) = \left[ A\cos\theta - B \right]_+. \tag{48}$$

When $A > B$, $U(\theta) = A\left[ \cos\theta - \cos\psi \right]_+$, which is non-zero only for $0 \le \theta \le \psi$. To get to the self-consistency conditions for the parameters, we go through procedures very similar to what we did for the ring. We ultimately get

$$\cos\psi = \frac{\alpha \int_0^\psi (\cos\theta' - \cos\psi) \sin\theta' d\theta'}{\int_0^\psi (\cos\theta' - \cos\psi) \cos\theta' \sin\theta' d\theta'} \tag{49}$$

and

$$\mu = \tfrac{1}{2} \int_0^\psi (\cos\theta' - \cos\psi) \cos\theta' \sin\theta' d\theta'. \tag{50}$$

Working out the integrals gives the parametric equations

$$\alpha = \tfrac{1}{3} \cos\psi (2 + \cos\psi), \tag{51}$$

$$\mu = \tfrac{1}{3} \sin^4 \frac{\psi}{2} (2 + \cos\psi). \tag{52}$$

Moving over to the case of the rotation group, for $g, g' \in SO(3)$ we take the $3 \times 3$ matrix representations $\mathbf{R}(g), \mathbf{R}(g')$ and consider $\tfrac{1}{3} \mathrm{Tr}\left( \mathbf{R}(g) \mathbf{R}(g')^\top \right)$ to be the similarity kernel.

$$\left[ \int_{SO(3)} \left\{ \tfrac{1}{3} \mathrm{Tr}\left( \mathbf{R}(g) \mathbf{R}(g')^\top \right) - \alpha \right\} u_\gamma(g') d^3 g' \right]_+ = \mu u_\gamma(g). \tag{53}$$

The Haar measure normalized so that $\int_{SO(3)} dg' = 1$. Once more, we can find a solution with $\gamma$ corresponding rotation group element $g_0$ where the response is maximum, i.e.,

$$u_{g_0}(\Omega) = \tfrac{A}{2} \left[ \mathrm{Tr}\left( \mathbf{R}(g_0)^\top \mathbf{R}(g) \right) - 2\cos\psi - 1 \right]_+, \tag{54}$$

with $\psi, \mu$ being determined by $\alpha$ through self-consistency equation. Since $\mathbf{R}(g_0)^\top \mathbf{R}(g) = \mathbf{R}(g_0^{-1} g)$ is a rotation, its trace is determined by the angle of rotation $\theta$. Let us work out the self-consistency for the special case where $g_0$ is the identity element. $u_\gamma(g) = U(\theta)$, where $\theta$ is associated with rotation angle of $g$: $\mathbf{R}(g)$ has the eigenvalues $1, e^{i\theta}, e^{-i\theta}$, corresponding to eigenvectors $\hat{\mathbf{e}}_1, \frac{1}{\sqrt{2}}(\hat{\mathbf{e}}_2 + i\hat{\mathbf{e}}_3), \frac{1}{\sqrt{2}}(\hat{\mathbf{e}}_2 - i\hat{\mathbf{e}}_3)$, respectively. Integrating over $g$ for the same $\theta$ corresponds to averaging over a random orthonormal basis $\{\hat{\mathbf{e}}_1, \hat{\mathbf{e}}_2, \hat{\mathbf{e}}_3\}$. Using that fact, we could argue that $\int_{SO(3)} \mathbf{R}(g')^\top u_\gamma(g') d^3 g'$ is proportional to the identity matrix.

Hence,

$$\mu U(\theta) = \int_{SO(3)} \left\{ \tfrac{1}{3} \operatorname{Tr}\left(\mathbf{R}(g)\mathbf{R}(g')^\top\right) - \alpha) \right\} u_\gamma(g') d^3 g'$$

$$= \tfrac{1}{3} \sum_{ij} \mathbf{R}(g)_{ij} \int_{SO(3)} \mathbf{R}(g')_{ij} u_\gamma(g') d^3 g' - \alpha \int_{SO(3)} u_\gamma(g') d^3 g'$$

$$= \tfrac{1}{3} \sum_{i} \mathbf{R}(g)_{ii} \int_{SO(3)} \mathbf{R}(g')_{ii} u_\gamma(g') d^3 g' - \alpha \int_{SO(3)} u_\gamma(g') d^3 g'$$

$$= \tfrac{1}{9} \operatorname{Tr}(\mathbf{R}(g)) \int_{SO(3)} \operatorname{Tr}(\mathbf{R}(g')) u_\gamma(g') d^3 g' - \alpha \int_{SO(3)} u_\gamma(g') d^3 g'$$

$$= \frac{1 + 2\cos\theta}{3} \int_0^\pi \frac{1 + 2\cos\theta'}{3} U(\theta')(1 - \cos\theta') \frac{d\theta'}{\pi} - \alpha \int_0^\pi U(\theta')(1 - \cos\theta') \frac{d\theta'}{\pi}. \quad (55)$$

The last step involves changing variables to $\theta'$ and getting the appropriate Jacobian $(1 - \cos\theta')$, before integrating out the variable related to the rotation axis. The factor $\frac{1}{\pi}$ is needed to maintain that $\int_{SO(3)} dg' = \int_0^\pi (1 - \cos\theta') \frac{d\theta'}{\pi} = 1$. Going through similar steps as before, and noting that $U(\theta') \propto [\cos\theta' - \cos\psi]_+$, we get the self-consistency conditions to be

$$1 + 2\cos\psi = \frac{9\alpha \int_0^\psi (\cos\theta' - \cos\psi)(1 - \cos\theta') d\theta'}{\int_0^\psi (\cos\theta' - \cos\psi)(1 + 2\cos\theta')(1 - \cos\theta') d\theta'}, \quad (56)$$

$$\mu = \frac{2}{9\pi} \int_0^\psi (\cos\theta' - \cos\psi)(1 + 2\cos\theta')(1 - \cos\theta') d\theta'. \quad (57)$$

Doing these integrals gives us the parametric equations for the $SO(3)$ problem to be

$$\alpha = \frac{(1 + 2\cos\psi)(6\psi - 3\sin\psi - 3\sin 2\psi + \sin 3\psi)}{27(-2\psi + 4\sin\psi + \sin 2\psi - 4\psi\cos\psi)}, \quad (58)$$

$$\mu = \frac{1}{54\pi}(6\psi - 3\sin\psi - 3\sin 2\psi + \sin 3\psi). \quad (59)$$

## A.7 Relation between NSM-1 and NSM-2 in Presence of Symmetry

Written in terms of $\mathbf{Q}$, without rank constraints, and with an inequality for $\operatorname{Tr}\mathbf{Q}$, NSM-2 is stated as

$$\min_{\mathbf{Q} \in \mathcal{CP}^T} -\operatorname{Tr}(\mathbf{DQ}) \quad \text{s.t.} \quad \mathbf{Q}\mathbf{1} = \mathbf{1}, \ \operatorname{Tr}(\mathbf{Q}) \le \beta T. \quad \text{(QNSM-2)}$$

Introducing Lagrange multipliers, the Lagrangian version of QNSM-2 becomes

$$\min_{\mathbf{Q} \in \mathcal{CP}^T} \max_{\boldsymbol{\alpha} \in R^T} \max_{\lambda \ge 0} -\operatorname{Tr}(\mathbf{DQ}) + \boldsymbol{\alpha}^\top(\mathbf{Q}\mathbf{1} - \mathbf{1}) + \lambda(\operatorname{Tr}(\mathbf{Q}) - \beta T). \quad (60)$$

When $\mathbf{D}$ is $G$-invariant, one can restrict the $\mathbf{Q}$ search to the $G$-invariant cone:

$$\min_{\mathbf{Q} \in \mathcal{CP}^G} \max_{\boldsymbol{\alpha} \in R^T} \max_{\lambda \ge 0} -\operatorname{Tr}(\mathbf{DQ}) + \boldsymbol{\alpha}^\top(\mathbf{Q}\mathbf{1} - \mathbf{1}) + \lambda(\operatorname{Tr}(\mathbf{Q}) - \beta T). \quad (61)$$

The $\boldsymbol{\alpha}$ dependence comes in through the term $\boldsymbol{\alpha}^\top \mathbf{Q}\mathbf{1}$. Since $\mathbf{Q} \in \mathcal{CP}^G$, we can play the trick of replacing $\mathbf{Q}$ by $\frac{1}{|G|} \sum_{g \in G} \mathbf{R}(g)\mathbf{Q}\mathbf{R}(g)^\top$.

$$\boldsymbol{\alpha}^\top \mathbf{Q}\mathbf{1} = \tfrac{1}{|G|} \sum_{g \in G} \boldsymbol{\alpha}^\top \mathbf{R}(g)\mathbf{Q}\mathbf{R}(g)^\top \mathbf{1} = \left( \tfrac{1}{|G|} \sum_{g \in G} \boldsymbol{\alpha}^\top \mathbf{R}(g) \right) \mathbf{Q}\mathbf{1}. \quad (62)$$

So we could replace $\boldsymbol{\alpha}$ by $\bar{\boldsymbol{\alpha}} = \frac{1}{|G|} \sum_{g \in G} \mathbf{R}(g^{-1})\boldsymbol{\alpha}$. When $G$ acts transitively, $\bar{\boldsymbol{\alpha}}$ is a constant vector, namely $\alpha\mathbf{1}$. In that case, the optimization problem becomes

$$\min_{\mathbf{Q} \in \mathcal{CP}^G} \max_{\alpha \in R} \max_{\lambda \ge 0} -\operatorname{Tr}(\mathbf{DQ}) + \alpha(\operatorname{Tr}(\mathbf{EQ}) - T) + \lambda(\operatorname{Tr}(\mathbf{Q}) - \beta T), \quad (63)$$

The necessary condition for $\mathbf{Y}$, where $\mathbf{Q} = \mathbf{Y}^\top \mathbf{Y}$, is now

$$[(\mathbf{D} - \alpha\mathbf{E})\mathbf{Y}^\top]_+ = \lambda \mathbf{Y}^\top, \quad (64)$$

which is the same equation we analyzed for $G$-invariant NSM-1.

## A.8 Derivation of the online algorithm

We start by changing the orders of optimization, and arriving at a dual of equation (13):

$$\min_{\mathbf{W}} \max_{\mathbf{b}} \sum_t l_t(\mathbf{W}, \mathbf{b}), \tag{65}$$

where

$$l_t(\mathbf{W}, \mathbf{b}) = \mathrm{Tr}(\mathbf{W}^\top \mathbf{W}) - \|\mathbf{b}\|_\mathbf{2}^\mathbf{2}$$
$$+ \min_{\mathbf{y}_t \geq 0} \max_{\mathbf{z}_t \geq 0} \max_{\mathbf{V}_t \geq 0} \left( -2\mathbf{x}_t \mathbf{W}^\top \mathbf{y}_t + 2\sqrt{\alpha}\mathbf{y}_t \cdot \mathbf{b} - \beta \|\mathbf{z}_t\|_2^2 + 2\mathbf{z}_t \mathbf{V}_t \mathbf{y}_t - \mathrm{Tr}(\mathbf{V}_t^\top \mathbf{V}_t) \right). \tag{66}$$

While this form is only an approximation to (13), it is convenient for online optimizaiton.

We solve (65) by an online alternating optimization of $l_t$ [44, 45]. First, we optimize with respect to $\mathbf{y}_t$, $\mathbf{z}_t$ and $\mathbf{V}_t$, by projected gradient descent-ascent-descent. This iteration results in the dynamics (14). Then, with optimal values of $\mathbf{y}_t$, $\mathbf{z}_t$ and $\mathbf{V}_t$ fixed, we do a gradient descent-ascent in $\mathbf{W}$ and $\mathbf{b}$. This results in (15).

# B   Additional NSM-2 Results

In figs. 8 to 10 we include a few additional examples that showcase the capabilities of NSM-2.

Figure 8: Manifold-tiling solutions of NSM-2: additional examples from synthetic datasets.

Figure 9: Manifold-tiling solutions of NSM-2: additional examples from the Yale Faces dataset. **From top to bottom:** using 2, 3, and 4 subjects, respectively. We build a 2d linear embedding (PCA) from the solution $\mathbf{Y}$ (each subject is identified by a different color in the right-most plot).

Figure 10: Manifold-tiling solutions of NSM-2: additional example with a synthetic 2d grid. We show a statistical summary of the receptive fields, after aligning them. **Bottom row:** A few representative receptive fields.

## Footnotes

[2] The proof of this statement relies on showing that if $\mathbf{Q}$ is an optimal solution then the group orbit average $\mathbf{Q}_0 = \frac{1}{|G|}\sum_{g \in G}\mathbf{R}(g)\mathbf{Q}\mathbf{R}(g)^\top$, an explicitly $G$-invariant matrix, has equal value of the objective function.