[Reviews · NeurIPS 2018]

Reviewer 1



SUMMARY OF THE PAPER This draft has a combination of good points and grey areas. The problem is of utmost importance and the research questions are clear, but the hypothesis (that neurons receptive fields are organized as low-dimensional manifolds expressed by atlases of locally overlapping charts) is presented as a fact. I’m not sure the normative approach from sociology does apply to neuroscience (see below). The material on the supplementary material provides hints about how to reach a proof but does not provide a proof for theorem 1. It is also said that they can show proof for Eq 9, but I’ve been unable to find a formal proof in the supplementary material. In general, I was left with the feeling of a good and intuitive idea to provide support for an interesting hypothesis, but with a rushed implementation with several gaps. STRONG POINTS • The problem, the research question and the hypothesis makes the potential interest and impact very high • The illustration that the manifold learning approach can have applications beyond neuroscience. • Results, despite my concerns about the formalities, are actually very good, suggesting that the authors intuition and heuristics is working pretty well WEAK POINTS • Some formal aspects are based on intuition and not formal derivation. • Methodological contribution is unclear; the objective functions is suggested to “have been proven successful” previously. • The underlying manifold traversing the observations is simply assumed to exist but never proven. Topological data analysis formally demonstrates that there are infinite topologies to a given cloud of points and it is a matter of choosing some topological constraint. Which one is chosen here and whether as a result, the observations lay in a manifold, is unclear. SUGGESTIONS TO (PERHAPS) IMPROVE THE PAPER Major • Personally, I find the use of the normative approach unnecessary and a long shot for this research. It would be easy to ask whether the normative approach has been validated for its use in neuroscience, but it is something more profound – normative analysis is about fitting expectations and not about decoding some biological truth from observations. If you put an objective function that guides a similarity-preserving mapping, it is no wonder you learn a similarity preserving manifold; that’s what you are fitting!. • Claims that the approach can learn any “arbitrary” manifold is not backed with any theoretical support. • How does the proposed model help to explain the opening problem regarding the understanding the localized receptive fields of neurons? Minor • The paper studies self organizing (tiling) manifolds and they seem to missed the self organizing maps (SOM). • Label all axis in all figures. • What do the authors mean by “unlike most existing manifold learning algorithms, ours can operate naturally in the online setting”.

Reviewer 2



This work proposes that biologically plausible representation of manifolds can be learned by optimizing objective functions that approximately preserve similarity of activities of upstream and downstream neurons. It analyzes the proposed objective function (NSM-1) and proposes an offline algorithm to optimize it. The paper then proceeds to propose an online algorithm that optimizes NSM-1 via a "biologically plausible" neural network. The simulations performed seemed to validate the underlying theory. Post rebuttal: I have read the rebuttal and other reviews. I will stay with my evaluation.

Reviewer 3



The ideas in this work are both promising and interesting and the general direction should be continued. The authors propose a novel biologically plausible framework on non-negative similarity-preserving mapping (NSM) and achieved manifold local tiling results. The arguments are further supported by some analysis results. If the following concerns can be reasonably addressed, this work should be considered as a paper for the conference. 1. Since manifold learning was initially proposed, most of the attention has been focused a parsimonious embedding guaranteed by both Whitney embedding theorem and Nash embedding theorem. A relatively higher dimensional embedding has been under appreciated. However, I suggest the author to further discuss the benefits of this practice and give more clear settings of these benefits. At the end the author mentioned "... the advantage of such high-dimensional representation becomes obvious if the output representation is used not for visualization but for further computation". This argument is not very satisfying and should be supported further in the discussion or experiments. 2. The connection between manifold tiling and manifold charting should be discussed. Building an atlas of a manifold could also be considered as a higher dimensional embedding. The following references should be cited: Brand 2003, Charting a manifold Pitelis et al. 2013, Learning a Manifold as an Atlas 3. The formulations throughout the whole paper changed quite a few times. I found this a little distracting and makes the central idea weaker. The authors should work on simplifying the formulations to make them more systematic and consistent. 4. The proof of in the supplement has a tendency to treat KKT conditions by using only part of the constraints, this is not safe. E.g. The Lagrangian in (20) should contain another term corresponds to the non negativity of Y. In general, to used the KKT condition for a constrained optimization problem, all the constraints should be considered at once. By only considering part of the constraints may lead to a false claim. I verified proposition 1 is correct, but certainly the argument is not very standard. In theorem 1, an addition closed cone constraint has been added. The authors should justify this new constraint doesn't break anything established in proposition 1. The result might be fine in this particular case. 5. The assumption for theorem 1 is quite strong. The author should provide more insights and explain the significance of this. If possible, please also provide some conjecture for a potential generalization. 6. Section 5 is very opaque and certainly not self-contained. It requires the reader to read another journal paper to make sense of it. If these update rules are not the contribution of this paper, it's better to put them into the supplement and discuss something more important. 7. In section 6, the third paragraph. Why is \beta T an upper bound of the rank? This is false without further constraints. 8. Under equation (21), "where the Lagrange multipliers ... must be adjusted so that ... sum to 1". Why? 9. Figure 4, the second subfigure from the right. The tiling doesn't seem very uniform. Is possible to provide an explanation to assist readers to understand further of the issue? 10. For reproducibility, I also encourage the authors to put a documented code online since the formulations and solution methods are quite complicated.

Reviewer 4



In this work, the authors show that similarity-preserving, the problem of creating an encoding of inputs on a manifold such that the similarity between encoded outputs between inputs matches similarity between inputs, yields localized receptive field that tile the space. They present a biologically plausible online neural learning model which combines several neural elements in a way that is equivalent to optimizing performance on this problem, and thus learns localized receptive fields. Additionally, they apply these methods to generate localized receptive fields over real-world datasets, such as MNIST and images of rotated objects. Overall I found this to be a good paper and enjoyed reading it. Quality: The paper starts from the basic, reasonable idea of similarity-preserving transformations, and then proceeds to prove mathematically, using sophisticated techniques, that the optimal receptive fields are localized, as well as show that these can be optimized by a biologically plausible learning rule. I view this as a significant result, although I don’t know the literature so well. The experimental results with MNIST and rotated objects are interesting, especially in that the manifold topology of inputs matches the manifold topology of encodings, but there is a somewhat preliminary feel to them, in that there is not much information about how well this scheme is able to generalize to unknown examples. Clarity: While I was not able to follow all proofs and mathematical detail, the authors use some fairly heavy-duty techniques, so I don’t think that’s a fair criticism. Given how dense the proof techniques are, I feel that the paper would be significantly improved if the authors took a more pedagogical/intuition-providing approach in general. For example, in Line 130, I have to admit I did not understand the proof of theorem 1, although it seems quite technical. I have some intuition for why this is true, but it might be helpful for the authors to provide some intuition. For example, there might be a much simpler an intuitive proof for the ring in the continuum limit; would it be possible for the authors to demonstrate this special case in a more pedagogical manner? line 184 Section 5: After staring for a little while, I saw how NSM-1 yields equation 11, but still don’t understand how the authors get from equation (11) to equation (12). The authors cite: Cengiz Pehlevan, Anirvan M Sengupta, and Dmitri B Chklovskii. Why do similarity matching objectives lead to hebbian/anti-hebbian networks? Neural computation, 30(1):84–124, 2018 nearby but there seems is no proof there either. Can the authors add a section in the supplemental proving that this is true and add more intuition behind what each of these terms are? Some intuition is provided in the main, but it is quite cursory given space limitations. During online optimization(line 188), what values were used for relative learning rates gamma and eta? My understanding of equations (13) (14), is that each step relies on a *single* sample x_t. Is this correct? If so, is the manner in which the manifold is explored important? For example, does it depend if x_{t}, x_{t+1} tend to be similar or unrelated? There seem to be other details missing, which is why I gave the current manuscript a medium reproducibility score rather than a high one. I suspect these details can be fixed and filled in without too much trouble. Originality and Significance: While I don’t claim to have a deep knowledge of the literature, this work seems to be original and significant. Receptive fields localized in physical/configuration space are found in many places, and understanding their function as well as their formation through the lens of similarity-preservation might be an important insight. There seem to be analytical results that exist concerning related similarity-preserving optimization problems, but not this particular one involving a cutoff where nearly-orthogonal inputs should yield completely-orthogonal encodings. Miscellaneous: In figure 4, how is the receptive field width and density (Y transpose, third matrix) determined by the input correlation structure (X transpose X, first matrix)? Can the authors provide some intuition behind this? It would also be nice if the authors provided some intuition/supplementary figures for how receptive fields depend on various parameters in the manner of SI figure 6.